# Characterization of controlling factors for soil organic carbon stocks in one Karst region of Southwest China

Qiang Li[1], Baoshan Chen[2], Hezhong Yuan[3], Hui Li[4], Shunyao Zhuang●[1]*

**1** State Key Laboratory of Soil and Sustainable Agriculture, Institute of Soil Science, Chinese Academy of Sciences, Nanjing, China, **2** School of Marine and Atmospheric Sciences, Stony Brook University, Stony Brook, NY, United States of America, **3** School of Environmental Science and Engineering, Nanjing University of Information Science & Technology, Nanjing, Jiangsu, China, **4** Department of Crop and Soil Sciences, North Carolina State University, Raleigh, NC, United States of America

* syzhuang@issas.ac.cn

**Data Availability Statement:** All relevant data are within the paper and its Supporting Information files.

**Funding:** The project was funded by the National Key R & D Project of China (2017YFD0800505). The funders had no role in study design, data

## Abstract

Soil organic carbon (SOC) contributes the most significant portion of carbon storage in the terrestrial ecosystem. The potential for variability in carbon losses from soil can lead to severe consequences such as climate change. While extensive studies have been conducted to characterize how land cover type, soil texture, and topography impact the distribution of SOC stocks across different ecosystems, little is known about in Karst Region. Here, we characterized SOC stocks with intensive sampling at the local scale (495 representative samples) via Random Forest Regression (RF) and Principal Component Analysis (PCA). Our findings revealed significant differences in SOC stock among land cover types, with croplands exhibiting the lowest SOC stocks, indicating that management practices could play a crucial role in SOC stocks. Conversely, there was little correlation between SOC stock and clay percentage, suggesting that soil texture was not a primary factor influencing SOC at a local scale. Further, Annual Precipitation was identified as the key driving factor for the dynamics of SOC stocks with the help of RF and PCA. A substantial SOC deficit was observed in most soils in this study, as evaluated by a SOC/clay ratio, indicating a significant potential in SOC sequestration with practical measures in the karst region. As such, future research focused on simulating SOC dynamics in the context of climate change should consider the controlling factors at a local scale and summarize them carefully during the upscaling process.

## Introduction

In terrestrial ecosystems, soil organic carbon (SOC) contributes to the largest carbon pool and plays a crucial role in global carbon cycling [1]. Research has suggested numerous factors affecting carbon accumulation and, consequently, the SOC distribution in ecosystems [2, 3]. Variations in SOC stock may induce dramatic changes in atmospheric $CO_2$ contents and lead to potential climate change in the systems with various scales [4–6]. Three main categories of

collection and analysis, decision to publish, or preparation of themanuscript.

**Competing interests:** The authors have declared that no competing interests exist.

factors have been identified to affect SOC stock dynamics in temporal and spatial scales: 1) climatic factors such as precipitation and temperature [7, 8]; 2) topographical parameters (*e.g.*, elevation and slope) [3, 9]; 3) soil physicochemical properties including soil texture and pH [10, 11]. However, most research has focused on one or two categories of factors separately, leading to considerable uncertainties in predicting SOC stock dynamics [8]. Therefore, there is a need to investigate the combined effect of these factors to better understand their role in regulating SOC dynamics in a particular scale of the ecosystem.

Previous research has suggested that soil texture, land cover, and their interactions are key factors affecting the dynamics of SOC stocks [12, 13]. For instance, long-term studies have demonstrated that the clay fraction of soil tends to accumulate more SOC than other fractions, particularly in wet regions where clay minerals are the primary regulators of SOC accumulation [14]. In contrast, studies conducted in semi-arid sandy regions have found weak associations between SOC and clay content [15]. However, their relationship can be complex and varies under different conditions [16], and the association at the local scale is still a subject of debate. Some studies have found no or little association between SOC and clay content at the local scale [17], while other factors aside from soil texture have been shown to impact SOC dynamics at a small scale [18]. Therefore, it is essential to investigate and evaluate the individual contributions of soil texture, land cover type, and climate at a specific tempo-spatial scale. Additionally, vegetation features can alter soil texture, which can complicate the elucidation of how SOC stocks respond to soil texture due to the interplay between soil texture and land cover type [19].

Compared to non-karst areas, Karst landforms have unique characteristics: i) thin soils; ii) low groundwater levels; iii) high water infiltration rate [20–24]. Karst landforms cover around 10% of the Earth's land surface and play a significant role in supplying water to human populations, particularly in southwest China, which contains one of the world's largest continuous karst areas. However, this region suffers from severe rocky desertification and is generally overpopulated and underdeveloped [19]. Previous studies have investigated the dynamics of soil organic carbon (SOC) in various regions worldwide [20, 24]. However, there is a dearth of research on karst-related SOC, particularly in karst agroecosystems. Given the high vulnerability of karst agroecosystems, it is crucial to gather information on SOC dynamics to improve our understanding of agro-environmental management and the carbon geochemical cycle process in this region.

The present study collected large quantities of representative samples from various fields in Bijie City, located in the sensitive karst regions of southern China, and classified them into three land cover types [25]. Then RF was used to figure out how soil texture, land cover type, and climatic variables affect SOC stock dynamics in the karst region along with other statistical methods. The findings of this study are valuable for mitigating climate change and land degradation, and promoting sustainable land management.

## Materials and methods

### Study area

Bijie City is located at the junction of Sichuan, Yunnan, and Guizhou provinces of China, which also belongs to subtropical humid climate zones with an annual mean temperature of 13°C and precipitation of 900–1400 mm, respectively. The elevation of Bijie City decreased from west (2885 m) to east (471 m). The eastern part of the city, such as Jinsha County, is suitable for agricultural development, with fragmented surfaces, hilly landforms, and water networks. The western part, like Weining County, is suitable for forestry and animal husbandry, characterized by high mountains, steep slopes, and large undulations. However, soil erosion is

a significant issue in this region. Tobacco-planting soil farming methods are standard in this region, mainly based on continuous cropping of flue-cured tobacco or irregular rotation of flue-cured tobacco with wheat, corn, potatoes, and rapeseed. The standard farming method for tobacco planting in this area is continuous cropping of flue-cured tobacco or irregular rotation of flue-cured tobacco with wheat, corn, potatoes, and rapeseed. Soil texture was identified as primarily silt loam according to the National Cooperative Soil Survey (web soil survey. cs. egov.usda.gov), with the predominant soil type of Utlisol (Soil Taxonomy) in this study.

### Soil sample collection and analysis

Between 2018 and 2019, a total of 495 representative sample points were gathered from tobacco fields in Bijie City, utilizing uniform spatial sampling and the tobacco planting situation as a guide (see Fig 1). The sampling locations were recorded by Global Positioning System (GPS) locator. At each location, three sets of surface soil samples (0–10 cm) were obtained using a 100 cm$^3$ cutting ring. The samples were labeled, packed in sealed bags, and then wrapped with plastic tape for storage.

The SOC content was measured using the potassium dichromate oxidation method [26]. Soil total nitrogen (TN) was measured with the Kjeldahl method [26]. SOC and TN (kg m$^{-2}$) stocks at depths of 0–10 cm were calculated as follows:

$$S = C * H * SBD / 100$$

where $S$ (kg m$^{-2}$) is SOC and TN stocks in the soil layer; $C$ (g kg$^{-1}$) is SOC and TN content in the soil layer; $H$ (cm) is soil layer thickness; and $SBD$ (g cm$^{-3}$) is the soil bulk density [27].

The soil texture was determined by measuring the volumetric percentages of sand (50–2000 μm), silt (2–50 μm) and clay (< 2 μm) fractions using a Laser scattering particle analyzer

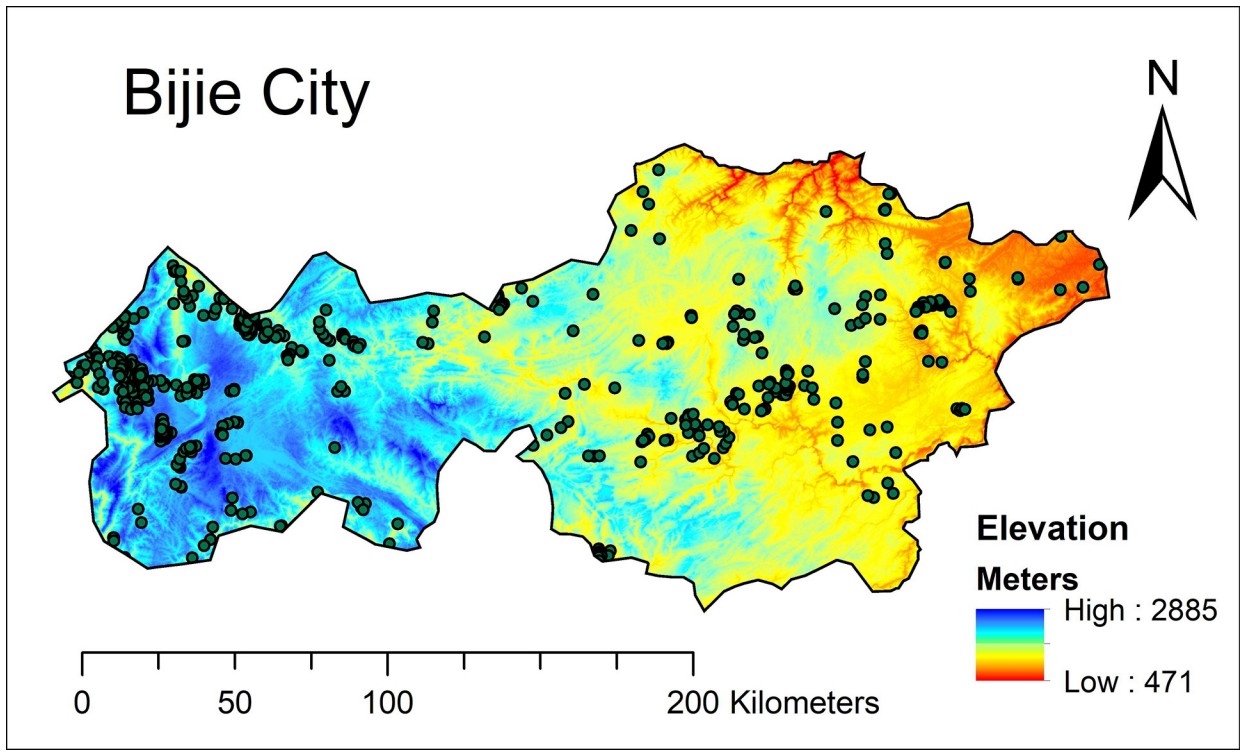

**Fig 1. Spatial distribution of sampling sites of SOC across Bijie City.**

(LS13320, Beckman Coulter, Inc.) based on laser diffraction technique [27]. Soil pH was measured by creating a 1:2.5 w/v soil-water suspension and using a pH meter. Soil bulk density (SBD) was determined by dividing the weight of the undisturbed soil cores by the container volume after drying the samples collected using a 100 cm$^3$ cutting ring at 105˚C until reaching a stable weight [27].

### Dataset construction

The climate data was sourced from Terra Climate with a 4-km spatial resolution [28]. The data were averaged over a period of 30 years (1991–2020) to obtain the annual accumulated precipitation and mean annual temperature. Topographic variables such as elevation were obtained from data with a spatial resolution of 30 m [29]. The land cover data was obtained from MODIS with a 500 m spatial resolution [25], which can be majorly divided into three land covers: savannas, tree cover 10–30% (canopy > 2m), and croplands.

### Data analysis

As a machine-learning technique, random forest regression (RF) was employed to identify the factors that influence the variation in SOC content among producer fields. In this study, the importance of independent variables was quantified by RF with Anaconda 3. The correlations between soil physicochemical properties, soil texture, and climatic factors were analyzed using RF at regional and local scales. Factors with high importance to SOC content were selected for the next step. Only factors with relative importance above or around 10% were selected. The Principal Component Analysis (PCA) was conducted in SPSS, while random forests were constructed using Anaconda 3. ANOVA and LSD test were used to determine if soil physicochemical properties were statistically different among the three land covers and the statistical significance was checked using Duncan's test at p = 0.05 via SPSS.

## Results

### Statistical description of variables

The details of soil texture characteristics under three land covers in Bijie City are shown in Table 1. The clay, silt, and sand percentages varied from 27.3% to 29.6%, 44.8% to 48.6%, and 21.8% to 27.4%, respectively, at a depth of 0–10 cm, with the highest values in croplands and the lowest values in savannas. The land cover type had a significant impact on soil texture (Table 1). Soils in crop/natural land had greater clay and silt content but lower sand content than those in savannas and croplands. In contrast, savannas and croplands had similar particle size fractions. Moreover, the average SBD (Table 1) decreased in the following order at a depth of 0–10 cm: croplands (1.22 g cm$^{-3}$) > savannas (1.19 g cm$^{-3}$) > crop/natural (1.14 g cm$^{-3}$).

The variations in SOC contents, SOC stock, TN contents, and TN stock among three land cover types are shown in Fig 2. As expected, SOC was found to be highly correlated with TN, with an $R^2$ value of 0.76, indicating a consistent C/N ratio at the regional scale. At the 0–10 cm

**Table 1. Volumetric contents of sand, silt, and clay of soils at different land cover types in our study area.**

| Land Cover | Clay (< 2 μm) | Silt (2–50 μm) | Sand (50–2000 μm) |
|---|---|---|---|
| Savannas | 27.93 ± 7.09 b | 44.84 ± 8.44 ab | 27.24 ± 11.24 a |
| Croplands | 27.32 ± 10.31 b | 45.26 ± 8.78 b | 27.42 ± 13.39 a |
| Crop/Natural | 29.60 ± 8.95 a | 48.57 ± 8.88 a | 21.83 ± 10.60 b |

Different letters indicate the mean values are significantly different at *p* = 0.05.

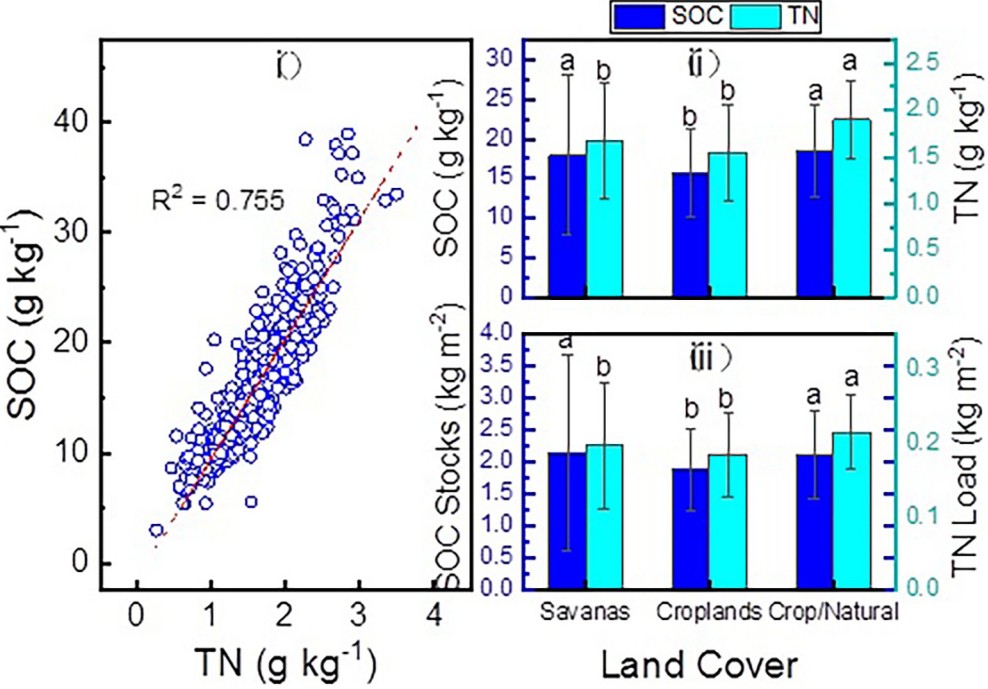

**Fig 2.** Relationship between SOC and TN content (i), SOC stock variations (ii), and TN (iii) at different land cover types. The dotted line in (i) represents the correlation between SOC and TN at the regional scale ($p < 0.05$).

depth, significant variations were observed in SOC and TN contents (Fig 2–ii) and stocks (Fig 2–iii) among the three land cover types. Croplands had the lowest SOC and TN contents (15.7 g kg⁻¹ and 1.55 g kg⁻¹, respectively), while savannas and crop/natural had higher SOC and TN contents (17.9 g kg⁻¹, 1.66 g kg⁻¹ and 18.5 g kg⁻¹, 1.90 g kg⁻¹, respectively). After considering the dynamics of bulk density, similar trends were observed in the variation of SOC and TN stocks, indicating that land cover played a significant role in SOC and TN distribution.

## Controlling factors on SOC contents with PCA and RF

At the regional scale, there was a close relationship ($p < 0.05$) between soil nutrients and texture at the regional scale. Changes in SOC stocks were found to be co-related to the clay content ($p < 0.05$), but the Principal Component Analysis (PCA) revealed that SOC stocks were not strongly correlated with either clay or silt content (Fig 3). Furthermore, neither elevation nor climatic factors such as temperature and precipitation appeared to have a noticeable effect on soil texture or SOC stock. At all sites, clay content was found to be negatively correlated with SBD.

The results of RF regression analysis showed a good fit for the prediction of SOC content, with a relative mean squared error (RMSE) of less than 20%. Fig 4 illustrates the relative importance of individual variables contributing to SOC content. The analysis identified three variables with importance greater than 10%: bulk density (22%), annual precipitation (11%), and altitude (11%). These findings suggest that bulk density, annual precipitation, and altitude, are important factors affecting the variation in SOC content among fields (Fig 4).

The negative correlation between SBD and SOC content was further confirmed, while a positive relationship between annual precipitation and SOC content was observed in all fields. Additionally, the low contribution (approximately 3%) of clay and silt content to the variation

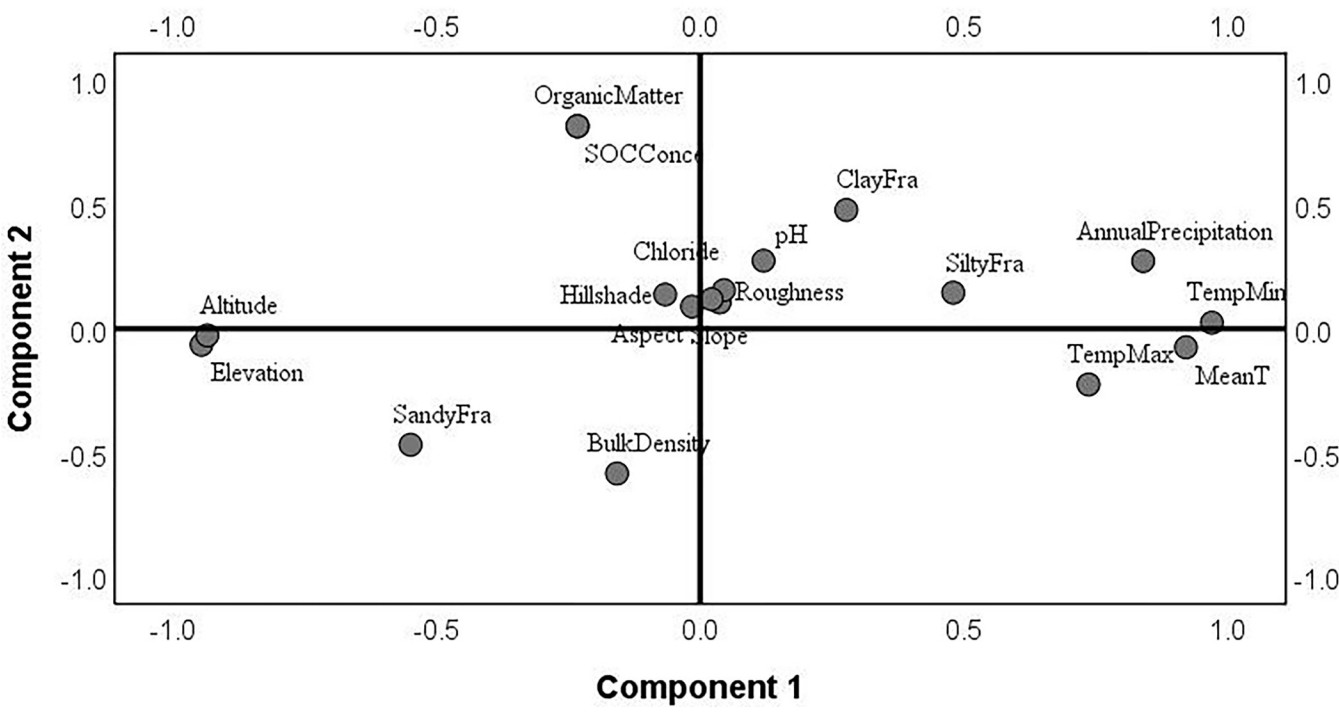

**Fig 3. Loading plot of soil and climatic indicators using PCA analysis.** Component 1 represents xx. Component 2 represents xx.

in SOC content, as revealed by the PCA results, suggests that soil texture is not a significant controlling factor in this study.

## SOC variation with clay contents and SOC/clay ratio

Fig 5 shows that the average SOC contents in croplands were lower than those in savannas and crop/natural soils, while the clay and silt contents were similar across the land cover types. The study employed SOC/clay ratios to indicate the structural condition of the soils, which were divided into "very good," "good," "moderate," and "degraded" levels based on thresholds of 1/8, 1/10, and 1/13 [30]. The results demonstrated that none of the soils in the study area were in the "very good" soil structure category, as their SOC/clay ratios were all below 1/8. The proportions of sites distributed among the three SOC/clay thresholds varied significantly among land cover types, particularly at the SOC/clay = 1/13 threshold (Fig 5–iii), where croplands had the lowest SOC/clay ratios. In fact, many sites in croplands had SOC/clay ratios smaller than 1/13 and even 1/10.

## Discussion

### Controlling factors for SOC via RF and PCA

Numerous studies have suggested that the dynamics of SOC are regulated by a complex interplay of factors, including soil conditions, topographic parameters, and climate [7, 31–33]. However, in this study, we found that the land cover type was the primary factor regulating SOC accumulation, indicating that land use and agricultural practices, such as cropping

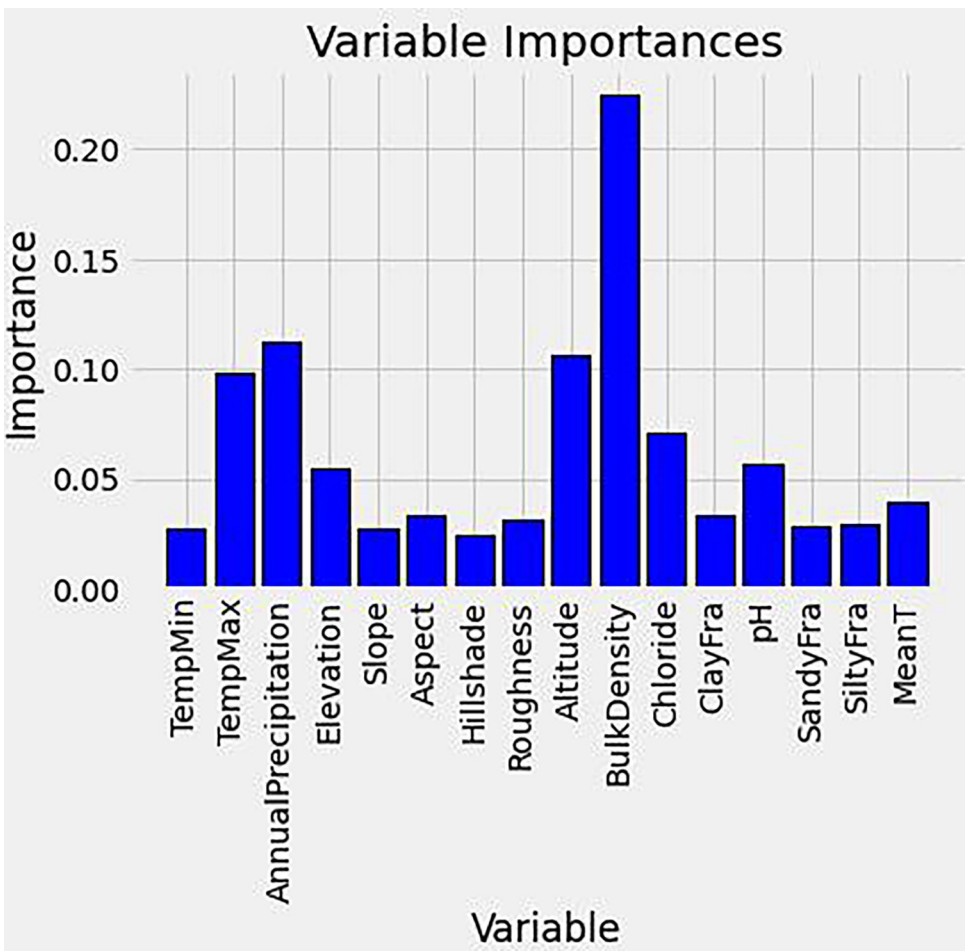

**Fig 4. Variable importance of soil and climatic factors influencing SOC contents using RF.** Y axis (blue column) represents and X axis are individual climatic factors. Blue column represents individual importance.

systems, fertilizer management, and tillage, may overshadow the effects of climate. These results are consistent with previous studies [34], which also found that land use exerted a significant impact on SOC in the Loess Plateau.

The regulation of SOC dynamics is a complex process involving several factors that work together [19]. Focusing on a single factor, such as land cover type, may lead to significant uncertainties and challenges in predicting SOC precisely [8, 35]. To overcome these limitations, RF and PCA were employed to identify the factors regulating SOC stocks in different land cover types (Figs 3 and 4) in this study. Not surprisingly, SBD can explain the highest portion of variability for SOC stocks because SOC is negative to the bulk density. Higher SOC content contributes to soil aggregates and better soil structure, which may lead to lower bulk density [36, 37]. In addition, SBD can directly determine the microbial processes involved in decomposition, and different soil conditions could lead to soil-specific stabilization or destabilization of SOC [38–41]. Another factor influencing SBD was the type of soil parent material, with carbonate rocks, argillaceous rocks, and quartzites associated with lower bulk density soil, and weathered red crust and sand shale associated with high bulk density soil [42, 43]. Meanwhile, management practices could be another critical factor affecting the variations of SBD. For example, deep vertical rotary tillage can reduce SBD by 0.11–0.12 g cm$^{-3}$ and improve soil

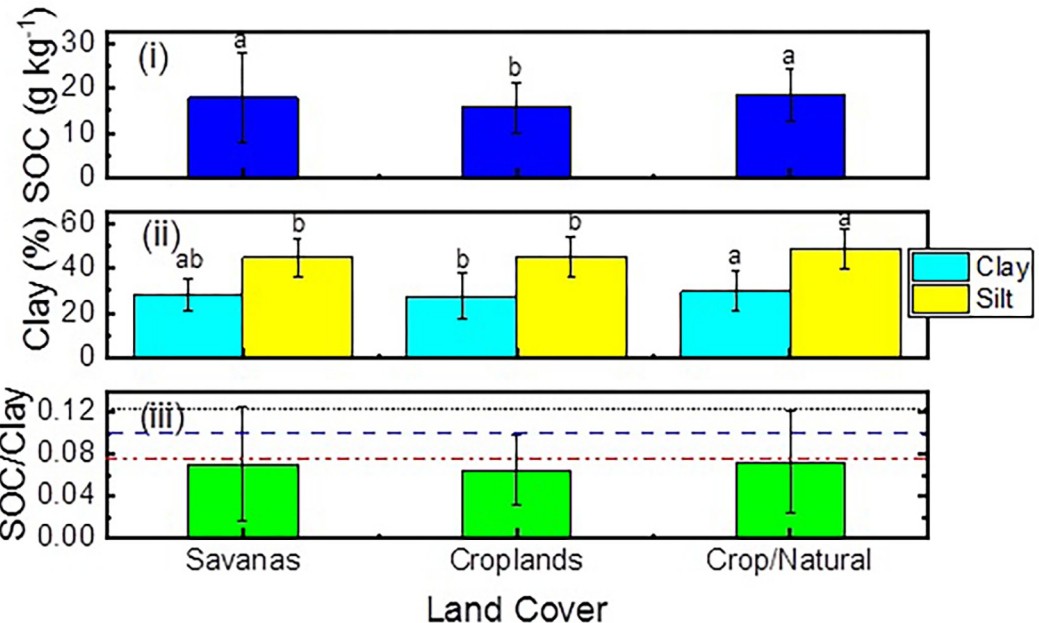

**Fig 5.** SOC content (i), clay content (ii), and SOC/clay ratio (iii) at different land cover types. Horizontal lines are SOC/clay thresholds: Short Dot = 1/8, Dash = 1/10, Dash Dot Dot = 1/13.

porosity from 4.66 to 6.81% [44]. A similar effect was observed for deep plowing in mountain tobacco fields [21].

Interestingly, followed by Altitude and Annual Temperature, Annual Precipitation was the most critical factor governing SOC stocks across land cover types in the Karst regions, since SBD is naturally linked to SOC. The state of SOC and physical quality is primarily influenced by climatic conditions within various bio-geographical zones in the terrestrial ecosystem [36, 45]. Our study confirms that there is a positive relationship between SOC stocks and clay content in areas with higher precipitation levels. This indicates that alterations in climatic conditions can affect the association between SOC and clay content. Precipitation and temperature likely play a role in this relationship, as they can lead to the weathering of soil minerals with more clay particles creation [46, 47]. Higher precipitation levels can also provide more water to soil, stimulating the growth of net primary productivity (NPP) [45, 48]. This can bring carbon into soils as root exudates and litter [49, 50]. Furthermore, a high clay content can result in more organic carbon molecules adsorbed onto clay surfaces due to the greater specific surface area and the more polyvalent cations. This creates organo-mineral complexes that help preserve SOC by inhibiting microbial and enzymatic decomposition, which in turn promotes SOC storage [14, 51, 52]. Our study demonstrates that climate change-induced increases in clay content can increase SOC storage in karst regions.

Compared to similar studies, the relatively low contribution of Annual Precipitation and Annual Temperature (while are still key factors) contributing to the SOC dynamics may result from the unique characteristics that distinguish karst from non-karst regions [20]. The occurrence of carbonate rocks intermixed with clastic rocks or situated between them is common in karst regions. These rocks show differences in their mineral composition, degree of weathering, and weathering mode. Unlike other catchments, precipitation in karst catchments quickly drains into underground systems via diverse fissures and conduits, resulting in varying water availability in soil profile and strong interactions between surface water and groundwater [47,

53, 54]. In contrast, an increase in precipitation in non-karst regions could retention soil water to a relatively high level, and would increase net NPP with sufficient water. Low water availability can influence vegetation distribution in landscape and weaken the influence on SOC stocks in karst regions [24, 55]. The actual SOC stocks in the karst-dominated area may be more sensitive to land-use change and less to precipitation, than that in the non-karst.

## Relationship between soil texture and SOC

Numerous studies have documented the impact of clay or clay+silt on soil organic carbon (SOC) dynamics. The role of clay in protecting SOM from microbial decomposition and increasing SOC stocks via adsorption and aggregation has been demonstrated by various studies [43, 53]. In contrast, silt has been found to be only partially reactive, with some carbon remaining unabsorbed [47, 56]. Differently as the widely reported, this study found a significantly weak relationship between clay and soil chemical characteristics that had little impact on SOC accumulation (Fig 5). The results suggest that vegetation characteristics, litter input, and root exudates, rather than clay content, under similar climatic conditions, may explain these trends (Figs 4 and 5). Other studies have also shown that vegetation features may be responsible for changes in clay contents at a local scale, with complex plant features such as canopy and residue cover decreasing soil erosion of fine particles and ultimately increasing clay content [15, 36, 57]. These findings are in agreement with previous studies that have shown that local-scale variables, such as soil properties, may account for a significant part of the variance in ecosystem processes, even at a plot scale, compared to climate [42, 58]. Furthermore, the study results indicated that SBD, as discussed in Section 4.1, had a more significant impact on SOC dynamics than clay content chemistry [15].

Many studies have stated soil nutrient dynamics, which is caused by changes in the quantity and quality of litter input, plays a crucial role in the dynamics of SOC. Our study found that SOC was significantly related to TN stocks (Fig 2–i), which is consistent with other similar studies [59–62], suggesting that nutrients play a critical role in driving SOC dynamics at a local scale, as expected [63]. In general, soil N levels regulate the microbial community structure, which is responsible for SOM decomposition [30]. The C is generally used for microbial growth rather than being released into the environment via respiration and extracellular enzyme excretion [64, 65]. This process could lead to a slower respiration rate and an increase in organic C accumulation as Annual Precipitation increases [66]. However, excessive N supply could have a negative impact on fungal growth, resulting in a decrease in SOC stocks. Not surprisingly, TN had a positive effect on SOC in topsoil under different land cover types in this study. One potential reason for this could be that the plant root distribution affect the vertical pattern of soil nutrients [28, 47], which subsequently affects SOC dynamics. Therefore, the significant synergistic effect of N could shape microbial community structure, influence SOM decomposition, and ultimately modulate SOC dynamics.

## Variation in SOC/clay ratio with land cover type

The ratio of SOC to clay is a useful indicator of soil physical conditions, as it combines information on both soil texture and organic matter content [30, 67]. In many soils, SOC can be bound to clay minerals such as allophonic minerals, 1:1 and 2:1 phyllosilicates [68, 69], leading SOC to be resistant to microbial decomposition. This high variability in protective capacity largely depends on soil type and ambient physio-chemical conditions [70–72]. Our study found that croplands had lower SOC/clay ratios and a higher proportion of sites with ratios below 1/10 compared to savannas and natural/croplands. This result was consistent with

previous studies [30, 67], indicating croplands may possess different soil physio-chemical characteristics compared to the other two land cover types.

Solid-state $^{13}$C nuclear magnetic resonance (NMR) spectroscopy of density fractions has been used to investigate the composition of occluded and non-occluded SOC and has found that the higher proportions of alkyl C in occluded SOC were more resistant to microbial decomposition compared to carbohydrates [73–75]. Our analysis showed that soil in Bijie City was depleted in SOC content compared to other natural systems, potentially due to the higher proportion of carboxyl C in acidic soils, which is more prone to microbial decomposition [76, 77]. Moreover, the predominant pH in this study was *ca*. 6.2, compared to occluded SOC, while the data for this part was unavailable. This finding is comparable to the finding of [29, 30], where soil fertility in the karst region was relatively low. Our finding showed that half of the croplands sites had SOC/clay <1/13, which supported the use of SOC/clay = 1/13 as a threshold for degradation evaluation, as savannas and crops/natural soils were not generally subject to significant disturbance and similar to semi-natural systems. Savannas soils were at intermediate degradation between cropland and crops/natural soils. The significant variation of SOC/clay ratio within each land cover and soil group demonstrated that clay content was not the only determinant of SOC dynamics, especially since land-use history had significant effects too. Under different land management, the thresholds are always different among soils despite the scatter [78]. A note of caution is the operational definition of the land cover to categorize SOC stocks based on the dataset of MODIS, which land cover data showed only land use around the sampling fields generally. This and any other difference in classification resulting in the difference of the driving factors for SOC stocks in this study area should be accounted for accordingly.

In the karst catchments, the changes in precipitation and potential evapotranspiration were more significant compared with the non-karst, with an actual evapotranspiration negatively along with the hydrological sensitivity [21, 79]. Additionally, the incorporation of $CO_2$ and the NPP are closely linked to biological transpiration [80, 81], which accounts for the major portion of evapotranspiration. Previous studies have highlighted that changes in land use and land cover can have a significant impact on factors such as runoff, groundwater recharge, and actual evapotranspiration [79, 82]. Therefore, regions dominated by karst landscapes are at a greater risk of experiencing degradation due to the impacts of climate change when compared to non-karst areas. Moreover, the karst-dominated region in China is facing challenges related to socio-economics, population, and environment [24, 83], and climate change is expected to exacerbate these issues. Therefore, policies must be implemented to enhance soil and water conservation and promote ecological protection in this region.

## Conclusions

Overall, our results highlight the weak relationship between SOC stocks and clays in the karst regions that supported by the combined effects of soil properties, precipitation, temperature, and land use. Croplands were found to have lower SOC stocks compared to savannas and natural/croplands, suggesting the need for improved management practices to increase SOC storage. While it is not surprising, Annual Precipitation was identified as one of the key driving factors for SOC dynamics in the karst agroecosystem. A SOC/clay ratio was used to evaluate soil conditions. Most soils in this study had a substantial SOC deficit, indicating a significant potential in SOC storage using carbon sequester measures in the karst region. In this study, the data available on SOC dynamics is expanded in amount in karst-dominated regions. Our results provide a preliminary landscape of SOC characteristics and potential controlling factors in the karst region; however, there is still considerable uncertainty, and more data are required

to present the SOC dynamics in this landform in the context of climate change. This research can have implications for management practices not only in the karst region but also in other regions worldwide.

## Supporting information

**S1 Data.**
(XLSX)

## Acknowledgments

We thank Dr. Yakun Zhang's hard work in extracting data from the USGS website. We also thank the reviewers for their highly constructive comments.

## Author Contributions

**Conceptualization:** Qiang Li, Shunyao Zhuang.

**Data curation:** Qiang Li, Baoshan Chen.

**Formal analysis:** Qiang Li, Hui Li.

**Methodology:** Baoshan Chen, Hezhong Yuan.

**Project administration:** Shunyao Zhuang.

**Writing – original draft:** Qiang Li.

**Writing – review & editing:** Shunyao Zhuang.

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
