## [Decision Letter · Decision Letter 0]

21 Jun 2023

PONE-D-23-13220Characterization of controlling factors for soil organic carbon stocks in the Karst region of Southwest ChinaPLOS ONE

Dear Dr. Zhuang,

Thank you for submitting your manuscript to PLOS ONE. After careful consideration, we feel that it has merit but does not fully meet PLOS ONE’s publication criteria as it currently stands. Therefore, we invite you to submit a revised version of the manuscript that addresses the points raised during the review process.

ACADEMIC EDITOR: The study has value but the manuscript has some problems as suggested by the three reviewers. The authors should respond to the comments of the reviewers one by one and revise the manuscript accordingly. The revised manuscript would be sent to the reviewers for further reviewing.==============================

We look forward to receiving your revised manuscript.

Kind regards,

Jian Liu

Academic Editor

PLOS ONE

“We thank Dr. Yakun Zhang's hard work in extracting data from the USGS website. We also thank the reviewers for their highly constructive comments. The project was funded by the National Key R & D Project of China (2017YFD0800505).”

“The project was funded by the National Key R & D Project of China (2017YFD0800505). The funders had no role in study design, data collection and analysis, decision to publish, or preparation of themanuscript.”

6. We note that Figure 1 in your submission contain [map/satellite] images which may be copyrighted. All PLOS content is published under the Creative Commons Attribution License (CC BY 4.0), which means that the manuscript, images, and Supporting Information files will be freely available online, and any third party is permitted to access, download, copy, distribute, and use these materials in any way, even commercially, with proper attribution. For these reasons, we cannot publish previously copyrighted maps or satellite images created using proprietary data, such as Google software (Google Maps, Street View, and Earth). For more information, see our copyright guidelines: http://journals.plos.org/plosone/s/licenses-and-copyright.

Reviewers' comments:

Reviewer's Responses to Questions

**Comments to the Author**

1. Is the manuscript technically sound, and do the data support the conclusions?

Reviewer #1: Partly

Reviewer #2: Yes

Reviewer #3: Yes

2. Has the statistical analysis been performed appropriately and rigorously? 

Reviewer #1: Yes

Reviewer #2: No

Reviewer #3: Yes

3. Have the authors made all data underlying the findings in their manuscript fully available?

Reviewer #1: Yes

Reviewer #2: No

Reviewer #3: Yes

4. Is the manuscript presented in an intelligible fashion and written in standard English?

Reviewer #1: Yes

Reviewer #2: No

Reviewer #3: Yes

5. Review Comments to the Author

Reviewer #1: The paper analyzed the relationship between SOC stock and the potential affecting factors in a Karst region in China. The soil sampling, soil quality analysis and data analysis were basically correctly conducted, however, the interpretation of the results is often quite confusing, mostly in the Discussion part. Some statements or arguments seemed to be contradictory to each other. Some examples are as following:

1. How is the Discussion part organized? The part one is discussing the controlling factors for SOC, but the second one is still discussing the relationship between SOC and soil texture and the third one is on SOC/clay ratio with land cover type. The structure of the discussion leads to many confusions.

2. In Line 176-177, ”These findings suggest that bulk density, annual precipitation, and altitude, are important factors affecting the variation in SOC content among fields (Fig 4)”. While in Line 206- 211 “However, in this study, we found that the land cover type was the primary factor regulating SOC accumulation, indicating that land use and agricultural practices, such as cropping systems, fertilizer management, and tillage, may overshadow the effects of climate”. These two statements seem to be contradictory to each other.

3. Line 265-267. How can we draw the conclusions that vegetation characteristics can explain the relationship between soil texture and SOC from the information in Figure 4 and Figure 5?

4. Line 304-305. What does “that soil in Bijie City was depleted in SOC content”? mean? There was no SOC in the soil?

5. In Line 263-265, “Differently as the widely reported, this study found a significantly weak relationship between clay and soil chemical characteristics that had little impact on SOC accumulation (Fig 5).” But in Line 336-338, the authors stated that “Overall, our results highlights the relationship between SOC stocks and clays in the karst regions that supported by the combined effects of soil properties, precipitation, temperature, and land use.” Are these two statements are contradictory to each other ?

6. Precipitation is important in Line 340-341, but not important in Line 244-245. Which statement is right?

Overall, there are so many confusing statements in the manuscript, which make readers hardly understand what is the right logic in this research.

Reviewer #2: This manuscript examines the controlling factors of soil organic carbon stocks in the Karst region of southwest China. The results will provide theoretical support for carbon neutral targets. However, there are some questions need to be improved in the manuscript, especially in the abstract, introduction, results, and discussion.

1. There was little correlation between soc stock and clay percentage, indicating that clay percentage is not a good way to characterize soc stock. On the contrary, SOC deficit in the manuscript was assessed by SOC/clay ratio, and the authors obtain results of significant SOC sequestration potential with practical measures in the Karst region. This makes me doubt the validity of the conclusion. The logical relationship between clay percentage and SOC sequestration potential should be further confirmed.

2. There were gaps between the results and conclusions. The conclusion of the manuscript was based on the results of soc stocks, while most of the results were focused on SOC contents, including random forest regression (Fig. 4), SOC variation with clay contents and SOC/clay ratio (Fig. 5). The conclusions should be supported by the results. Thus, the authors should clarify the relationship between results and conclusions.

3. There is a section dedicated to the application of machine learning analysis to the manuscript. That would be fine if it was a manuscript focused on statistical methods. Otherwise, it is not necessary to introduce statistical methods detailed in the introduction (Line 56–63).

4. The discussion is not consistent with the conclusions. the discussion should be focus on the controlling factors for SOC, while the authors take too much time to discuss the factors for SBD (soil bulk density) (Line 219–227).

5. All pictures need to be modified according to the requirements of the paper. The picture should be complete, clearly marked, and self-explanatory. Figure legend should be marked especially for Fig. 3 and Fig. 4. In addition, the proportion of interpretation for the first and the second axis should be marked.

6. All “p” should be italic. The “p” value should be provided for the relationship between SOC and TN in Fig.2.

7. Please check the citation carefully in the text, for example, in Line 193, Line 210. The family name of the cited paper is missing, and the serial number is marked in the upper right corner of the author’s name.

8. There are too many references in the manuscript. Please control the quantity.

Reviewer #3: This is a very interesting study. The method is relatively novel, and the conclusion is also very meaningful.

I have the following suggestions for this paper:

1. Can the region selected for this study represent the Karst region of Southwest China? The author need to further refine in the title.

2. There is no introduction to sampling in areas such as Savanas in the research methods.

6. PLOS authors have the option to publish the peer review history of their article (what does this mean?). If published, this will include your full peer review and any attached files.

Reviewer #1: No

Reviewer #2: No

Reviewer #3: No

---

## [Author Response · Author response to Decision Letter 0]

18 Aug 2023

Dear editors and reviewers:

We would like to express our appreciation to you and the reviewers for the critical feedback concerning our manuscript entitled “Characterization of controlling factors for soil organic carbon stocks in the Karst region of Southwest China” (ID: PONE-D-23-13220). We feel so lucky that our manuscript went to these reviewers as the valuable comments from them not only helped us with the improvement of our manuscript, but also suggested some useful ideas for future studies. Based on the comments we received, careful modifications have been made to the manuscript. All changes were marked in red text. We hope the new manuscript will meet all the necessary requirement for your magazine's standard. Below you will find our point-by-point responses to the editor and reviewer's comments/questions (red font refers to editors and reviewers' original comments).

Sincerely,

Dr. Shunyao Zhuang

The corresponding author

Response: Please check this part.

Response: This is a collaborating project. They have the authority to access the field sites.

“We thank Dr. Yakun Zhang's hard work in extracting data from the USGS website. We also thank the reviewers for their highly constructive comments. The project was funded by the National Key R & D Project of China (2017YFD0800505).”

“The project was funded by the National Key R & D Project of China (2017YFD0800505). The funders had no role in study design, data collection and analysis, decision to publish, or preparation of the manuscript.” Please include your amended statements within your cover letter; we will change the online submission form on your behalf.

Response: Thanks for reviewer's excellent comment. Revision has been made on the revised manuscript about the funding information. Please check the competing interests and funding items.

Data Availability Statement: The relevant data can be available from the corresponding author.

Funding: The authors would like to thank financial support from the National Key R & D Project of China (2017YFD0800505). The funders had no role in study design, data collection and analysis, decision to publish, or preparation of the manuscript.

Competing Interests: The authors have declared that no competing interests exist.

Response: Many thanks for reviewer's excellent suggestions. There are no restrictions for the data in this manuscript, which contain two components of data. The first data is the public database shared by USGS (United States Geological Survey), which has been clearly stated that everyone can access them as long as giving necessary credit such as citating them clearly in the reference. The other part of data were collected by this project. Restrictions are not applicable for this part since the task is to publish and share this part of the data.

Response: Thanks for the reviewer's good suggestion. Revision has been made on the revised manuscript.

6. We note that Figure 1 in your submission contain [map/satellite] images which may be copyrighted. All PLOS content is published under the Creative Commons Attribution License (CC BY 4.0), which means that the manuscript, images, and Supporting Information files will be freely available online, and any third party is permitted to access, download, copy, distribute, and use these materials in any way, even commercially, with proper attribution. For these reasons, we cannot publish previously copyrighted maps or satellite images created using proprietary data, such as Google software (Google Maps, Street View, and Earth). For more information, see our copyright guidelines: http://journals.plos.org/plosone/s/licenses-and-copyright.

Response: Figure 1 was made using ArcGIS by us authors instead of being copied and pasted from other sources. No permissions are needed.

Reviewers' comments:

Reviewer's Responses to Questions

Reviewer #1: The paper analyzed the relationship between SOC stock and the potential affecting factors in a Karst region in China. The soil sampling, soil quality analysis and data analysis were basically correctly conducted, however, the interpretation of the results is often quite confusing, mostly in the Discussion part. Some statements or arguments seemed to be contradictory to each other. Some examples are as following:

1. How is the Discussion part organized? The part one is discussing the controlling factors for SOC, but the second one is still discussing the relationship between SOC and soil texture and the third one is on SOC/clay ratio with land cover type. The structure of the discussion leads to many confusions.

Response: We apologize for this confusion. Thanks for reviewer's great comment. The logic is as follows: firstly, run the general analysis (such as random forest) like other studies. However, no consistent pattern (such as the relationship between SOC and soil texture) was observed. Then discussed the potential reason why. The third step is to establish relationships between SOC and combined parameters such as SOC/clay ratio. The last part is to propose that the unique characteristics of Karst Region may be the cause of peculiar observations between SOC and other factors proved by similar studies.

2. In Line 176-177, ”These findings suggest that bulk density, annual precipitation, and altitude, are important factors affecting the variation in SOC content among fields (Fig 4)”. While in Line 206- 211 “However, in this study, we found that the land cover type was the primary factor regulating SOC accumulation, indicating that land use and agricultural practices, such as cropping systems, fertilizer management, and tillage, may overshadow the effects of climate”. These two statements seem to be contradictory to each other.

Response: Sorry for this confusion. In our opinion, there are no contradictory statements. The first part of the statement results from the summary of similar research related to this topic. The second part of the statement comes from our project. Due to the unique characteristics of the Karst region in this research, we have some different conclusions, which justify the necessity for this project and are also the novelty of this research. 

3. Line 265-267. How can we draw the conclusions that vegetation characteristics can explain the relationship between soil texture and SOC from the information in Figure 4 and Figure 5?

Response: Sorry for this confusion. I think our intention is to point out that different land cover types have significantly different SOC, implying vegetation characteristics maybe the cause of SOC (Figure 5). Meanwhile, soil texture can explain small portion of SOC dynamics based on Figure 4. 

4. Line 304-305. What does “that soil in Bijie City was depleted in SOC content”? mean? There was no SOC in the soil?

Response: We apologize for this confusion. This sentence means that SOC content are relatively low in Bijie City compared to similar research.

5. In Line 263-265, “Differently as the widely reported, this study found a significantly weak relationship between clay and soil chemical characteristics that had little impact on SOC accumulation (Fig 5).” But in Line 336-338, the authors stated that “Overall, our results highlights the relationship between SOC stocks and clays in the karst regions that supported by the combined effects of soil properties, precipitation, temperature, and land use.” Are these two statements are contradictory to each other?

Response: Thanks so much for the reviewer's careful observation. They are contradictory to each other. Revision has been made on the revised manuscript from line 339 to 340.

6. Precipitation is important in Line 340-341, but not important in Line 244-245. Which statement is right?

Response: Thanks for pointing it out. Revision has been made on the revised manuscript from line 240 to 245.

Overall, there are so many confusing statements in the manuscript, which make readers hardly understand what is the right logic in this research.

Reviewer #2: This manuscript examines the controlling factors of soil organic carbon stocks in the Karst region of southwest China. The results will provide theoretical support for carbon neutral targets. However, there are some questions need to be improved in the manuscript, especially in the abstract, introduction, results, and discussion.

1. There was little correlation between soc stock and clay percentage, indicating that clay percentage is not a good way to characterize soc stock. On the contrary, SOC deficit in the manuscript was assessed by SOC/clay ratio, and the authors obtain results of significant SOC sequestration potential with practical measures in the Karst region. This makes me doubt the validity of the conclusion. The logical relationship between clay percentage and SOC sequestration potential should be further confirmed.

Response: We apologize for this confusion. The logic is as follows: firstly, run the general analysis (such as random forest) like other studies. However, no consistent pattern (such as the relationship between SOC and soil texture) were observed. Then discussed the potential reason why. The third step is to establish relationships between SOC and combined parameters such as SOC/clay ratio. The last part is to propose that the unique characteristics of Karst Region may be the cause of peculiar observations between SOC and other factors proved by similar studies. We admit that there was a mistake on the conclusion, which has been corrected.

2. There were gaps between the results and conclusions. The conclusion of the manuscript was based on the results of soc stocks, while most of the results were focused on SOC contents, including random forest regression (Fig. 4), SOC variation with clay contents and SOC/clay ratio (Fig. 5). The conclusions should be supported by the results. Thus, the authors should clarify the relationship between results and conclusions.

Response: Thanks for this excellent comment. We admit that SOC contents cannot be equivalent to SOC dynamics. However, SOC stocks are linearly related to SOC contents (has been clarified from Line 99 to Line 104). The factors associated with SOC contents will affect SOC stock eventually.

3. There is a section dedicated to the application of machine learning analysis to the manuscript. That would be fine if it was a manuscript focused on statistical methods. Otherwise, it is not necessary to introduce statistical methods detailed in the introduction (Line 56–63).

Response: Thanks for this great suggestion. Revision has been made in the Introduction part of the revised manuscript.

4. The discussion is not consistent with the conclusions. the discussion should be focus on the controlling factors for SOC, while the authors take too much time to discuss the factors for SBD (soil bulk density) (Line 219–227).

Response: Thanks for pointing it out. SBD is also another factor for SOC stocks along with SOC content (from line 99 to line 104). Please check the response to Comment #1 and #2. 

5. All pictures need to be modified according to the requirements of the paper. The picture should be complete, clearly marked, and self-explanatory. Figure legend should be marked especially for Fig. 3 and Fig. 4. In addition, the proportion of interpretation for the first and the second axis should be marked.

Response: Thanks for reviewer's excellent comment. Revision has been made on the figures of revised manuscript. Please check and add something to explain x and y axises of Fig 3 and Fig. 4.

6. All “p” should be italic. The “p” value should be provided for the relationship between SOC and TN in Fig.2.

Response: Thanks for reviewer's excellent suggestion. Revision has been made on the revised manuscript.

7. Please check the citation carefully in the text, for example, in Line 193, Line 210. The family name of the cited paper is missing, and the serial number is marked in the upper right corner of the author’s name.

Response: Thanks so much for these excellent suggestions. Revision has been made to the revised manuscript about the citations.

8. There are too many references in the manuscript. Please control the quantity.

Response: Many thanks for reviewer's excellent comment. References have been deleted to some extent.

Reviewer #3: This is a very interesting study. The method is relatively novel, and the conclusion is also very meaningful.

I have the following suggestions for this paper:

1. Can the region selected for this study represent the Karst region of Southwest China? The author need to further refine in the title.

Response: Thanks for the reviewer's good suggestion. Revision has been made on the revised manuscript about the title.

2. There is no introduction to sampling in areas such as Savanas in the research methods.

Response: Many thanks for reviewer's excellent comment. Sorry for this confusion. All the samplings are the same for the 495 field sites (method section). The dataset was then majorly divided into three land covers: savannas, tree cover 10–30% (canopy > 2m), and croplands artificially.

---

## [Decision Letter · Decision Letter 1]

18 Dec 2023

Characterization of controlling factors for soil organic carbon stocks in the Karst region of Southwest China

PONE-D-23-13220R1

Dear Dr. Zhuang,

We’re pleased to inform you that your manuscript has been judged scientifically suitable for publication and will be formally accepted for publication once it meets all outstanding technical requirements.

Kind regards,

Jian Liu

Academic Editor

PLOS ONE

Additional Editor Comments (optional):

All comments have been addressed.

Reviewers' comments:

Reviewer's Responses to Questions

**Comments to the Author**

1. If the authors have adequately addressed your comments raised in a previous round of review and you feel that this manuscript is now acceptable for publication, you may indicate that here to bypass the “Comments to the Author” section, enter your conflict of interest statement in the “Confidential to Editor” section, and submit your "Accept" recommendation.

Reviewer #1: All comments have been addressed

Reviewer #2: (No Response)

2. Is the manuscript technically sound, and do the data support the conclusions?

Reviewer #1: Yes

Reviewer #2: (No Response)

3. Has the statistical analysis been performed appropriately and rigorously? 

Reviewer #1: Yes

Reviewer #2: (No Response)

4. Have the authors made all data underlying the findings in their manuscript fully available?

Reviewer #1: Yes

Reviewer #2: (No Response)

5. Is the manuscript presented in an intelligible fashion and written in standard English?

Reviewer #1: Yes

Reviewer #2: (No Response)

6. Review Comments to the Author

Reviewer #1: The authors have addressed all my questions and revised the MS. and now the manuscript can be accepted.

Reviewer #2: (No Response)

7. PLOS authors have the option to publish the peer review history of their article (what does this mean?). If published, this will include your full peer review and any attached files.

Reviewer #1: No

Reviewer #2: No

---

## [Editor Report · Acceptance letter]

15 Jan 2024

PONE-D-23-13220R1 

PLOS ONE

Dear Dr. Zhuang, 

I'm pleased to inform you that your manuscript has been deemed suitable for publication in PLOS ONE. Congratulations! Your manuscript is now being handed over to our production team.

Kind regards, 

on behalf of

Dr. Jian Liu 

Academic Editor

PLOS ONE